# Avian iPSC Derivation to Recover Threatened Wild Species: A Comprehensive Review in Light of Well-Established Protocols

**DOI:** 10.3390/ani14020220

**Published:** 2024-01-10

**Authors:** Iara Pastor Martins Nogueira, Guilherme Mattos Jardim Costa, Samyra Maria dos Santos Nassif Lacerda

**Affiliations:** Laboratory of Cellular Biology, Department of Morphology, Institute of Biological Sciences, Federal University of Minas Gerais, Belo Horizonte 31270-901, MG, Brazil; iarapnvet@gmail.com (I.P.M.N.); gmjc@ufmg.br (G.M.J.C.)

**Keywords:** induced pluripotent stem cells, bird, feather follicle cells, somatic reprogramming, germ cell, conservation

## Abstract

**Simple Summary:**

Imagine that you are an adult with a specific job, such as a biology teacher. Now, suppose it were possible for you to return to your childhood with your entire future ahead of you and endless possibilities of jobs that you can pursue. This review thoroughly describes all the steps of how cells can undergo this process. We compare different techniques that are already well established to transform cells that previously served a specific function in the organism into stem cells, which can give rise to any other type of cell. For instance, we explain how it is possible to convert bird feather cells into induced pluripotent stem cells. Once these cells can be differentiated into any other type, with additional effort, they can be directed towards gamete formation, meaning eggs and sperm. This technique represents an important strategy for the future conservation of endangered avian species, whether through the establishment of biobanks or in breeding programs.

**Abstract:**

Induced pluripotent stem cells (iPSCs) were first generated by Yamanaka in 2006, revolutionizing research by overcoming limitations imposed by the use of embryonic stem cells. In terms of the conservation of endangered species, iPSC technology presents itself as a viable alternative for the manipulation of target genetics without compromising specimens. Although iPSCs have been successfully generated for various species, their application in nonmammalian species, particularly avian species, requires further in-depth investigation to cover the diversity of wild species at risk and their different protocol requirements. This study aims to provide an overview of the workflow for iPSC induction, comparing well-established protocols in humans and mice with the limited information available for avian species. Here, we discuss the somatic cell sources to be reprogrammed, genetic factors, delivery methods, enhancers, a brief history of achievements in avian iPSC derivation, the main approaches for iPSC characterization, and the future perspectives and challenges for the field. By examining the current protocols and state-of-the-art techniques employed in iPSC generation, we seek to contribute to the development of efficient and species-specific iPSC methodologies for at-risk avian species. The advancement of iPSC technology holds great promise for achieving in vitro germline competency and, consequently, addressing reproductive challenges in endangered species, providing valuable tools for basic research, bird genetic preservation and rescue, and the establishment of cryobanks for future conservation efforts.

## 1. Introduction

According to the 2021 IUCN Red List, approximately 11,000 bird species are facing varying degrees of threat and risk to their populations [1]. Numerous projects worldwide aim to conserve these populations through measures based on environmental education, social mobilization, preservation and restoration of natural habitats, specimen monitoring, and in situ reproductive assistance [2,3,4,5]. However, in the face of animal trafficking, habitat destruction due to direct and indirect human actions, emerging diseases, natural disasters, and challenging reproductive rates, there remains a prevailing negative pressure on at-risk populations [2,5,6].

The pursuit of convenient assisted reproductive technology (ART) is driven by the intention to provide a genuine opportunity for these avian species to overcome the rate of destruction and effectively recover their populations [7]. There are two major cornerstones for ex situ species conservation: the requirements that populations be self-sustainable and that they maintain a degree of genetic diversity [8]. Approaches such as the use of gamete cryopreservation, which is already a bottleneck for avian female reproduction [9,10], and laboratory-dependent techniques such as in vitro fertilization (IVF) and even avian cloning do not fulfill these proposed basic principles [11]. Moreover, stem cell-based technologies, such as the utilization of embryonic stem cells (ESCs) or adult multipotent stem cells, also necessitate invasive collection procedures. The acquisition of these cells puts at risk the viability of embryos or even specimens from endangered populations, rendering it impractical considering conservation goals [12,13]. In addition to all that, the biosafety of reproductive techniques is a critical consideration, since the ultimate objective in conservation is the recovery of populations in their natural habitats. For instance, the somatic cell nuclear transfer (SCNT) method may raise scientific and ethical concerns due to the potential risks of abnormal offspring development, thereby hindering its suitability for conserving wild animals [8,14].

In this scenario, the possibility of utilizing induced pluripotent stem cells (iPSCs) arises. First generated by Yamanaka in 2006 [15], iPSCs possess the capacity for self-renewal, exhibit a normal karyotype, and acquire pluripotency, allowing them to differentiate into cells from all three germ layers both in vitro and in vivo [15,16,17,18]. With numerous established protocols available, iPSCs can be generated reliably in both human and mouse models. and their applications have proven to be invaluable in a wide range of research areas, encompassing toxicology, pharmacological science, and regenerative medicine. iPSC biotechnology holds great promise in overcoming the challenges of traditional stem cell therapies, such as generating patient-specific cell lines, offering an ethical and practical alternative to address the dilemmas associated with the use of ESCs [18,19,20].

From the perspective of animal reproductive science, iPSCs have the potential to fulfill a pivotal role in the conservation of endangered species [8,12]. They provide a means to preserve the genetic material of the threatened species without the need to sacrifice individuals by enabling the reprogramming of adult cells from noninvasively collected tissues [12]. Moreover, as they are self-renewable, in theory, they offer an inexhaustible source of target cells, ensuring a sustainable supply of genetic material [8]. Equally important, the use of iPSCs allows us to access the diploid genome rather than just half of the alleles as provided by gamete cryopreservation [11,21]. Remarkably, iPSCs possess the demonstrated capability to undergo differentiation into primordial germline cells (PGCs) [22]. This property holds the potential to address the issue of low reproductive rates in some wild avian species, as it enables the production of avian gametes through xenotransplantation to produce germline chimeras [12,23,24]. This opportunity in iPSC technology provides a foundation for the development of innovative in vitro breeding programs, offering a potential solution to the challenges faced by endangered species in terms of reproduction efficiency and population recovery. 

iPSCs have already been successfully generated in some endangered wildlife species, primarily mammals, e.g., the Bengal tiger (*Panthera tigris*) [25] and northern white rhinoceros (*Ceratotherium simum cottoni*) [26], but also avian species such as the Okinawa rail (*Hypotaenidia okinawae*), Japanese ptarmigan (*Lagopus muta japonica*), and Blakiston’s fish owl (*Bubo blakistoni*) [27]. In general, it has been observed that the iPSC induction process is relatively conserved across different taxa [28,29]. However, a significant challenge lies in the fact that different species may have distinct requirements and demand specific techniques for achieving complete reprogramming and generating high-quality iPSCs [8,27]. Therefore, it is of great importance to extensively study the variations and adapt and optimize the iPSC derivation protocols for wild species, especially nonmammalian species. Herein, we discuss in detail the methods of cell harvesting, the most studied reprogramming factors, the delivery methods, and the molecular adjuvants used to generate and maintain avian iPSCs. Furthermore, we show the potential applications of using these cells (Figure 1). To establish a comparative analysis, the methodologies employed for human and mouse iPSC generation were utilized, as they offer a broader range of options and have well-established and replicable protocols. By examining and comparing these methodologies, this review intends to provide insights into and highlight potential strategies for improving iPSC generation in avian species, paving the way for future advancements in avian reproductive science and conservation efforts.

## 2. Somatic Cell Sources for Reprogramming

In avian species, although the entire embryo originates from the center of the epiblast and possesses remarkable regenerative abilities [13], manipulating a fertilized egg from an endangered species poses the risk of permanent contamination and definitive loss. Therefore, the utilization of pluripotent embryonic cells, such as chicken embryonic stem cells (cESCs) and chicken embryonic fibroblasts (CEFs), is deemed impractical for animal conservation research [12,30]. Additionally, due to the limited availability of post-hatch eggshells from wild species, avian amniotic stem cells, also known as amniotic fluid stem cells (AFSCs), may not be the optimal alternative for such studies [13].

In chickens, mesenchymal stem cells (cMSCs) have been isolated from various organs such as the liver, lung, and bone marrow. These cells have demonstrated the ability to differentiate into a wide range of cell types such as osteogenic, adipogenic, and endothelial [13,30,31]. However, collecting these cells in a noninvasive and feasible manner from healthy animals within their natural habitat also poses significant challenges. 

Thus, the capacity to generate pluripotent cells via the genetic reprogramming of adult somatic cells has brought about a revolutionary shift in stem cell research, not only exerting a substantial impact on the fields of regenerative medicine and drug discovery but also providing a valuable instrument for conducting research using avian models. Notably, iPSCs opened new avenues for the exploration and implementation of ex situ reproduction projects [12,13,30,31].

The first crucial criterion for achieving successful reprogramming is determining the appropriate source of a somatic cell type for inducing pluripotency once it significantly influences efficiency, kinetics, and the quality of iPSC generation. Other important aspects to consider when selecting the somatic cell type include cell availability, the process of cell acquisition, the requirements for cell maintenance, and its reprogramming capacity, ensuring that the chosen cell type aligns with the specific research objectives [32]. Consequently, numerous research groups have investigated diverse somatic cell sources in order to identify promising candidates for reprogramming into iPSCs. These sources primarily encompass easily accessible adult cells that can be isolated noninvasively or minimally invasively, such as dermal fibroblasts and keratinocytes, urine cells, and peripheral blood mononuclear cells (PBMCs) [19]. 

In the case of mice, the initial reprogramming studies were performed using mouse embryonic fibroblasts (MEFs) and tail-tip fibroblasts (TTFs) [15], and subsequent studies successfully achieved reprogramming of adult human dermal fibroblasts (HDFs) [16,33]. Regarding avian iPSC production from somatic tissues, fibroblasts have been extensively used due to their abundance, ease of cultivation, and well-defined protocols [13]. Some research groups achieved successful derivation of avian iPSCs from embryonic fibroblasts [23,24,30,31]. However, as previously mentioned, it is crucial to ensure that the obtainment of donor cells does not compromise the viability of the individual specimen when considering their potential application in conservation efforts. 

As an alternative, dermal fibroblasts have already been used to produce avian iPSCs [27,34,35]. Feathers are complex epidermal appendages that contain a pulp with a rich quantity of feather follicle cells (FFCs) of various types such as erythrocytes, keratinocytes, and dermal fibroblasts [34,36], which are one of the most recruited cell sources in avian iPSC derivation projects. These cells can be collected minimally invasively from birds by plucking feathers from the primary and secondary wing remiges [24,27,36]. Cell acquisition from the calamus pulp opening offers a simplified process for isolating skin cells through enzymatic and mechanical digestion, resulting in a high yield. Significantly, the quantity of tissue within the follicle exhibits variation based on the developmental stage of the feather, with nascent feathers exhibiting a greater abundance of pulp tissue [36]. Therefore, FFC obtainment in adult birds is performed in two steps. The first stage involves removing a mature feather, and after 15 to 20 days of first removal, a second assembly involves removing the reactivated calamus from the same feather, now containing a substantial amount of pulp tissue [36]. In a natural habitat, capturing the same bird consecutively presents a considerable challenge if not an impossibility. Therefore, an alternative approach involves collecting feathers from juvenile individuals that are still in the nest and undergoing the feathering phase, as they possess an actively developing pulp [24,36], potentially allowing for a single collection session to be sufficient. Another significant source of donor cells for reprogramming is dermal fibroblasts obtained from cutaneous tissue of recently deceased wild animals. Although under nonideal conditions, these cells demonstrated successful utilization in the generation of iPSCs of rare avian species [27]. As a potential strategy, these materials could also be collected in controlled environments, such as zoos during medical assessments [8], ensuring better sample quality and minimizing the risk of contamination.

In human medicine, fibroblasts have certain drawbacks that need to be considered when using them as a somatic cell source for generating human iPSCs (hiPSCs). These include the need for a skin biopsy, which can be undesirable; decreased reprogramming efficiency in aged patients; heterogeneity within fibroblast populations; and the risk of accumulating mutations due to their constant exposure to the environment [37,38]. This has led to research for alternative cell sources that meet specific criteria for biomedical applications. Keratinocytes have emerged as a promising cell source for hiPSC generation due to their accessibility. They can be alternatively obtained from hair follicles using noninvasive methods such as plucking a hair in the active growth phase. Reprogramming human keratinocytes to pluripotency has been achieved using retroviral transduction of Yamanaka factors (*OCT4*, *SOX2*, *KLF4*, and *c-MYC* - OSKM), resulting in higher efficiency, faster kinetics, and fewer retroviral integrations compared to fibroblast reprogramming due to their epithelial gene signature [19,37,39]. However, there are limitations to their use, such as fast senescence after a few passages and longer doubling time, requiring careful cultivation. 

Xi and colleagues (2013), reported that keratinocytes obtained from the internal epidermal layer of the feather follicle of adult White Leghorn chickens exhibit characteristics of multipotent stem cells. Chicken keratinocytes can be separated in vitro from the FFC fibroblasts, as the latter quickly attach to culture plates within 2 h, while keratinocytes take about 24 h to attach [34,36]. These cells were shown to be highly efficient in incorporating foreign genes, surpassing fibroblast-like cells, and displaying elevated levels of exogenous gene expression [34], highlighting the importance of feather keratinocytes in enabling genetic engineering techniques for endangered avian species. 

In addition to the use of keratinocytes for human iPSCs, urine cells offer a noninvasive means of collection that does not require professional assistance, making it readily accessible compared to other cell sources. Although urine cells consist of a heterogeneous population, successful reprogramming into hiPSCs has been achieved using both integrating and nonintegrating methods [40]. They naturally express stem cell-specific genes and surface markers associated with pluripotency, and their epithelial nature does not require them to undergo the mesenchymal-to-epithelial transition (MET) during reprogramming, leading to enhanced efficiency and kinetics [19,40,41]. Nevertheless, generating avian iPSCs from urine cells has never been accomplished. A plausible explanation for not using these cells in birds may be that it is impracticable to collect the urine of wild animals, as it would require the scientist to be present at the exact moment of excretion. Additionally, the avian urinary system shares the same pathway as the digestive system [42], raising concerns about potential sample contamination with intestinal content and microorganisms.

Another major source of hiPSCs are the peripheral blood mononuclear cells (PBMCs), which consist of various cell types, including lymphocytes, monocytes, dendritic cells, natural killer cells, and hematopoietic stem cells/progenitor cells. They offer several advantages as a somatic cell source for generating hiPSCs as they are highly proliferative and reprogrammable using viral-based techniques and integration-free approaches [19,43,44]. At present, there are no reports of avian PBMC-derived iPSCs. Since it is considered to be minimally invasive, blood collection from zoo animals or research animals could be used in tests for bird iPSC derivation. It would be of interest to investigate the behavioral characteristics of avian nucleated erythrocytes during the reprogramming process. 

Many somatic cell types have yet to be investigated for avian iPSC generation. Thus, future research should aim to validate the existence of more efficient and viable cellular sources for conservation applications in endangered species, surpassing the fibroblasts used by most research groups. Divergent physiological characteristics between avian and mammalian species manifest across several key parameters. Avian cells exhibit a significantly lower cellular basal metabolism, attributable to a concomitant reduction in mitochondrial lipid content and oxygen consumption. Additionally, avian cells demonstrate lower membrane polyunsaturation and diminished total antioxidant capacity compared to their mammalian counterparts [45]. These features may bear noteworthy implications for the reprogramming of primary somatic cells into iPSCs. The cellular metabolic rate has the potential to alter the kinetics and velocity of the reprogramming process, while disparities in oxidative stress parameters, such as lower basal cellular oxygen consumption and reduced lipid oxidative damage in avian cells, could impact the redox state and overall stress response of cells, thereby influencing the success of iPSC generation. Significantly, long-lived species in both avian and mammalian groups present intriguing distinctions, with fibroblasts from avian species exhibiting delayed yet prolonged phosphorylation of ERK [46], which could interfere not only in the acquisition of a pluripotent state but also in its maintenance.

Despite these observations, there is a notable absence of studies comparing the efficiency of the reprogramming process and the quality of iPSCs generated from avian, murine, and human somatic cells. Compounding this gap in knowledge are substantial differences, such as the higher core temperatures exhibited by avian species (ranging between 34 and 44 °C) in contrast to their mammalian counterparts (with core temperatures ranging between 30 °C and 40 °C), despite being subjected to similar cultivation conditions [47].

For chicken cells, research groups have expanded fibroblasts in Kuwana’s modified avian culture medium-1 (KAV-1) containing chicken serum [27]. However, despite cellular differences in metabolism, primary cultures of fibroblasts from mice, humans, and birds are carried out in a remarkably similar manner, utilizing Dulbecco’s modified Eagle’s medium (DMEM) with 10% fetal bovine serum (FBS) at an incubation temperature of 37 degrees Celsius and 5% CO_2_ [16,22,35,48,49]. Acknowledging these physiological nuances is imperative for refining reprogramming strategies and ensuring the optimization of culture conditions and growth factors specific to avian cells. The intricate interplay between avian-specific physiological traits and the reprogramming process warrants further exploration to advance the field of avian iPSC research. 

## 3. Reprogramming Factors

Yamanaka and colleagues successfully reprogrammed somatic cells of mice and humans based on the overexpression of the four defined transcription factors: *Oct3/4* (also known as *Pou5f1*), *Sox2*, *Klf4*, and *c-Myc* [15,16,17,49,50]. Rosselló and colleagues (2013) demonstrated that human OSKM genes are capable of deriving iPSCs not only in vertebrates, e.g., Galliformes, but also in invertebrates, providing evidence for the conservation of reprogramming pathways between different species [8,28]. 

The regulatory circuitry among these factors exhibits interconnectivity and feed-forward regulation. In ESCs, *Oct4*, *Sox2*, and *Klf4* were found to autoregulate, as the latter was observed to enhance the role of *Oct4* and *Sox2* in developmental pathways. *c-Myc* did not exhibit such a pattern as it is not autoregulated and plays a distinct role in metabolic processes of cells undergoing reprogramming [17,51,52]. Yamanaka factors have the capability to regulate well-established ESCs’ pluripotency-associated signaling pathways, including p53, Wnt, TGF-b, Hedgehog, and MAPK pathways, through various combinations [16]. Both *Oct4* (octamer-binding transcription factor 4) and *Sox2* (sex-determining region Y-box 2) have overlapping functions in the regulation of gene expression networks associated with a role of pluripotency maintenance. Dysregulation of these gene expressions leads to loss of pluripotency and cell differentiation [51,53]. The mouse *Oct4* gene was identified in chickens as *cPouV*, showing similarities to mammalian *OCT4* orthologues [51]. 

*Klf4* (Kruppel-like factor 4) was found to be involved in direct somatic reprogramming of fibroblasts into iPSCs, specifically in regulating pluripotency-associated genes such as *Klf2* and *Klf5* [31,51,54]. *c-Myc*, on the other hand, is considered a nonessential factor in iPSC generation, as in its absence, iPSCs can still be generated. However, *c-Myc* expression is essential for high-quality iPSCs due to its role in controlling histone acetylation [52,55].

In a complementary manner, certain genes, particularly *Nanog* and *Lin28*, were identified as enhancers for improving the efficiency of iPSC reprogramming in humans and birds [17,53,56,57]. *Nanog* (Nanog Homeobox) is a transcriptional factor, controlled by *Oct4* and *Sox2* in rewiring transcriptional networks to promote self-renewal and suppress differentiation by finely modulating epigenetic remodeling. Once *Nanog* is inactivated, the iPSCs differentiate into endoderm-like cells [56,57,58]. *Lin28* encodes an RNA-binding protein that regulates the let-7 family of microRNA, which controls genes related to differentiation and growth. Thus, *Lin28* plays an important role not only in early embryo development but also in regulating reprogramming, naive/primed pluripotency, and stem cell metabolism [59,60]. One of the pioneering studies in domestic birds demonstrated successful reprogramming of avian embryonic fibroblasts transduced with six human reprogramming factors: *POU5F1*, *NANOG*, *SOX2*, *LIN28*, *c-MYC*, and *KLF4* [31]. 

A major challenge in generating iPSCs from wild avian species is the need for specific reprogramming genes, hampering the standardization of the induction protocol. A recent study revealed a significant demand for reprogramming wild avian FFCs. A transposon vector carrying eight mouse reprogramming factors (*Oct3/4*, *Sox2*, *Klf4*, *c-Myc*, *Klf2*, *Lin28*, *Nanog*, and *Yap*) had to be employed to achieve a full pattern of reprogrammed Japanese golden eagle cells [27]. Activation of *Yap/Taz* was implicated in the reprogramming of mouse somatic cells into tissue-specific stem/progenitor cells as well as in the reprogramming of human adult cells into iPSCs [61]. *Yap* (yes1-associated transcriptional regulator) plays a significant role in the cellular expansion, self-renewal, and maintenance of the stem cell phenotype by participating in metabolic rearrangement to meet the changes in bioenergetic and biosynthetic demands [61,62].

*Yap* assumes a role marked by complexity and controversy. Despite its recognized promotion of stemness in various stem cell types, including pluripotent stem cells, its impact extends to the early determination of cell fate and differentiation, concurrently opposing pluripotency during the initial stages of embryogenesis [63,64,65]. This dual modality contributes to the intricate and occasionally contradictory nature of *Yap* engagement in pluripotency regulation, wherein it inhibits pluripotency induction autonomously within cells while concurrently fostering it in a non-cell-autonomous manner through microenvironmental alterations [64]. Given these complexities, further investigation into the specific effects of *Yap* in avian reprogramming processes would be valuable for a comprehensive understanding of its role in diverse biological contexts.

As a result, research efforts pertaining to each target species must be conducted in a unique and individual manner. This approach will ensure that species-specific considerations are considered, optimizing the success and efficiency of high-quality iPSC generation for each particular avian species.

## 4. Delivery Methods

The field of iPSCs can be considered still relatively nascent, and it continues to hold promise for new discoveries and advancements [20]. In general, the generation of iPSCs follows a straightforward concept: the induction of the ectopic expression of a combination of stem cell reprogramming factors followed by the confirmation of cellular dedifferentiation [15,16,17,48,50]. Currently, there are multiple strategies available for delivering reprogramming factors into the target cell [32,66,67]. However, when considering the application of iPSCs in endangered wildlife species, it is crucial to prioritize the biosafety of the techniques.

The original approaches employed in humans and mice utilized integrative retroviral vectors, such as pMXs or pMSCV, which primarily targeted dividing cells to deliver the Yamanaka factors [15,16,17,50]. Next, lentiviruses emerged as widely employed delivery vehicles for expressing reprogramming factors in somatic cells due to their ability to infect both dividing and nondividing cells [68,69]. Integrative viruses have the remarkable capability to incorporate their genetic material, typically RNA, into the host cell genome, thus becoming a permanent component of the cell’s genetic composition [70]. For that, they exhibit a high transduction efficiency that enables stable inheritance of the viral genes by the host cell and their transmission to subsequent daughter cells during cell division [68,69,70,71]. Lentiviruses have been implicated in the induction process of the first nonmammalian iPSCs in birds, specifically derived from quail embryonic fibroblasts. These iPSCs exhibited a significantly shorter doubling time compared to the original quail parent line, indicating enhanced proliferative potential and clonal expansion capabilities [30,31]. Lentiviruses also present a significant performance in reprogramming chicken cells, with about 1.5% of cells exhibiting positivity for pluripotent markers after 21 days [23].

In addition to potential barriers, such as immune response induction, scalability, and cost, the utilization of integrative vectors has raised several safety concerns for subsequent applications of iPSCs. These vectors rely on the use of potentially harmful viral particles that can lead to oncogene expression, posing a risk to the cells and their genomic stability. Furthermore, lentiviral vectors have the largest genomic footprint among iPSC generating methods, primarily due to the risk of insertional mutagenesis [13,69,71]. The random integration of these vectors into the host cell genome also leads to the generation of heterogeneous iPSC cell lines, which can hinder comparisons between different lines. 

To address concerns related to retrovirus integration, several silencing strategies can be implemented. Designing self-inactivating vectors is one approach, aiming to restrict viral gene expression postintegration and thereby mitigate the risk of unintended consequences [72]. Another strategy is integration site selection, which involves choosing vectors with preferences for integrating into genomic regions less likely to cause disruptions [73]. Epigenetic silencing is a complementary method that utilizes modifications like DNA methylation and histone modifications to repress retroviral sequences within the host genome [74]. Additionally, excision systems play a crucial role by employing site-specific recombinases or genome editing tools to precisely eliminate integrated retroviral sequences, providing a means to eradicate viral remnants [75]. Collectively, these diverse approaches contribute to the development of induced pluripotent stem cells (iPSCs) with enhanced safety profiles. However, the poor comprehension of how these strategies work on avian cells renders the viral integrative vectors not ideal for iPSC generation, especially when considering their potential use in breeding programs of endangered species aiming at reintroduction into the wild [70,76]. 

One alternative that has been explored in mammals to enable the utilization of lentiviruses is the development of Cre-lox lentiviral vectors. This approach involves using a standard lentiviral vector but incorporating loxP sites flanking the inserted transgenes. By transiently expressing Cre recombinase, the inserted transgenes can be efficiently deleted [77]. However, even with this approach, lentiviral vectors are still not considered safe for therapeutic or reproductive applications [70]. 

Nonintegrative viruses, such as adenoviruses, are also referred to as episomal viruses due to their characteristic of not integrating their genetic material into the host cell’s genome. Instead, these viruses replicate autonomously within both dividing and nondividing cells in the host cell’s nucleus as separate entities. The episomal DNA derived from these viruses can persist and be actively expressed for a limited period; however, it is not stably inherited by the host cell during subsequent cell divisions [70,78]. One drawback associated with the use of nonintegrative viruses is their lower efficiency in generating iPSCs compared to retroviruses [67,70,78]. For instance, the reprogramming efficiency exhibited several orders of magnitude lower, estimated to be around 0.001% to 0.0001% in mouse cells and 0.0002% in human cells [78]. Despite adenoviral vectors being considered a safer option for expressing reprogramming factors, the low efficiency is a limiting point given the scarcity and difficulty of access to cells from avian target genetics (Stadtfeld et al., 2008 [70]; Zhou and Freed, 2009 [78]). 

One notable nonintegrative strategy is the Sendai virus, an RNA virus that has demonstrated the ability to reprogram various mammalian cell types, including human fibroblasts and blood cells, with efficiencies of approximately 0.1% for blood cells and 1% for fibroblasts, surpassing the reprogramming efficiency of adenoviruses [79,80,81]. However, a challenge associated with the use of Sendai virus is its replication competence, making it difficult to completely eliminate the virus from all cells, even after multiple passages [79,80,81]. Despite most studies in avian species utilizing viral vectors for cell reprogramming [23,24,28,30,31], this delivery approach poses a major challenge for future applications of iPSCs regarding biosafety concerns about the genomic integration, which could hinder the reintroduction of these animals into their natural habitats. 

Exploring nonviral reprogramming methods, the PiggyBac transposon system may offer a promising alternative. This mobile genetic element utilizes a “cut and paste” mechanism, facilitated by the transposase enzyme, to excise the transposon from one genomic location and integrate it into another [82]. Notably, transposons demonstrate efficient and precise integration of large DNA segments into the genome [32,35]. In mouse cells, reprogramming using PiggyBac vectors has achieved efficiencies ranging from 0.02% to 0.05% [82]. Two efficient polycistronic transposon-based expression systems, PB-R6F (*Oct3/4*, *Sox2*, *Klf4*, *c-Myc*, *Klf2*, *Lin28*, and *Nanog*) and PB-TAD-7F (*Yap* added), were successfully utilized for iPSC induction from chicken fibroblasts [27,32,35]. One challenge associated with the use of PiggyBac for iPSC generation is the possibility of genomic alterations at the transposon insertion site, emphasizing the need for sequence verification [82]. Additionally, longer DNA sequences can pose difficulties in the efficient delivery and transposition process [82].

Another interesting option for iPSC generation is the delivery of episomal plasmids, which offers simplicity of implementation and eliminates the need for labor-intensive virus production and costs. Theoretically, this method provides a nonintegrating approach, resulting in the generation of footprint-free iPSCs [83,84]. However, there are challenges associated with this method. Multiple transfections make it challenging to control the dose of plasmid delivered to the cells throughout the reprogramming process. Furthermore, plasmids tend to be diluted faster in actively dividing cells. It is important to consider that transfection efficiency varies depending on the cell type, and larger plasmids generally exhibit lower transfection rates [83,84]. 

Reprogramming plasmids can also be delivered to target cells through other nonviral methods, such as chemical transfection using calcium phosphate or lipid vectors like liposomes, as well as physical transfection techniques like electroporation, microinjection, or the use of ballistic particles [43,84]. Nonetheless, the transfection rates are generally low, resulting in only a small fraction of cells incorporating and expressing the plasmid DNA. Moreover, some methods can cause cytotoxic effects, reducing cell viability. Optimization of transfection conditions, including adjusting reagent concentrations, timing, and choice of method, is crucial to enhance efficiency and minimize cytotoxicity. Ongoing research aims to develop more efficient and less toxic transfection techniques for improved delivery of plasmid DNA and enhanced success in reprogramming experiments.

Minicircle vectors, resembling miniplasmids, have also shown promising applications in successfully reprogramming cells to iPSCs not only in humans but also in chickens [76,85]. These vectors, composed of essential genetic elements, have a compact size that contributes to increasing transfection efficiencies, particularly advantageous when working with avian cells characterized by low transfection efficiency [85]. Moreover, the technology promoted prolonged gene expression compared to traditional plasmids, attributed to reduced activation of DNA-silencing mechanisms [76]. The minicircle reprogramming approach was investigated in chicken embryonic fibroblasts for the delivery of human reprogramming factors (*POU5F1*, *SOX2*, *NANOG*, and *LIN28*). Generated iPSCs exhibited high proliferation rates and demonstrated developmental plasticity by successfully differentiating into cells representative of all three germ layers, both in vitro and in vivo [85]. 

Nonintegrative reprogramming methods that do not involve the delivery of exogenous DNA have demonstrated significant potential in the realm of reprogramming. One such approach is mRNA transfection, which offers a footprint-free method for introducing reprogramming factors into cells. This method utilizes in vitro transcribed RNAs that can be introduced into cells using a cationic vehicle [86,87]. Efforts have been made to optimize mRNA transfection, resulting in reprogramming efficiencies ranging from 1.4% to 4.4% in human fibroblasts [86,87,88]. Although this method has the potential to increase induction efficiency in avian cells, it can be labor-intensive and requires daily mRNA transfections, resulting in increased cellular stress and severe cytotoxicity [32]. An alternative to synthetic mRNA-based reprogramming is the use of self-replicating RNA (srRNA), which enables an extended duration of protein expression [86], warranting investigation and evaluation in avian cells.

Further research and optimization are needed to fully explore the potential of these methods in avian reprogramming studies.

## 5. Reprogramming Enhancers

Reprogramming efficiency can vary significantly, even when employing the same delivery method. Small molecules have been identified that, once added to the cell medium, can enhance reprogramming results by targeting various molecular mechanisms, such as histone deacetylation, signaling pathways like TGFβ and MEK, epigenetic modifiers, the ROCK pathway, and glycolysis induction (Figure 2). 

Valproic acid and sodium butyrate, classified as histone deacetylase inhibitors, are commonly used in reprogramming protocols [89,90]. Senescence is a barrier to iPSC reprogramming, and valproic acid treatment can assist in pluripotency-related gene activation, the suppression of the p16/p21 pathway, and the alleviation of the G2/M phase blockage. Thus, it can mitigate senescence, enhance cell growth, and increase the generation of iPSC colonies [89,90,91]. In combination with valproic acid, Vitamin C, an antioxidant, promotes pluripotency gene transcription via DNA demethylation or altering histone modification, which facilitates the transition of pre-iPSC colonies to a fully reprogrammed state and alleviates cell senescence during iPSC generation [91,92,93]. Indeed, vitamin C was shown to decrease p53-p21 signaling and act as an agonist for specific histone demethylases [92,93,94,95].

One great reprogramming adjuvant is CHIR99021, a well-known selective glycogen synthase kinase-3 (GSK-3β) inhibitor. The timing and concentration of CHIR99021 affect various cell behaviors of iPSCs, including colony formation, cell proliferation, and differentiation [27,91,96]. Carefully designed CHIR treatment allows for enhanced proliferation of iPSCs without deviating from the undifferentiated state, as evidenced by the expression levels of pluripotency-associated genes [96].

Y-27632 is a Rho kinase inhibitor largely used in mammalian iPSC cultivation to control dissociation-induced apoptosis [97], support undifferentiated growth of pluripotent stem cells [98], and improve the recovery after cryopreservation [99]. Thiazovivin is a small molecule that also targets the Rho/ROCK pathway, protecting human ESCs in the absence of an extracellular matrix by regulating cell adhesion pathways such as E-cadherin, which mediates cell–cell interaction [91,100,101]. It promotes cell survival, maintains self-renewal capacity, and, in combination with inhibitors of the TGFβ receptor and MEK pathway, can improve reprogramming efficiency by over 200 times [100]. 

One option to use in combination with Y-27632 or thiazovivin is PD0325901, a reprogramming enhancer that acts as a mitogen-activated protein kinase (MAPK) inhibitor, preventing ERK activation and the subsequent phosphorylation of downstream targets involved in cell differentiation, favoring a pluripotent state [91,102,103]. MEK inhibitors also facilitate mesenchymal-to-epithelial transition (MET) during reprogramming by promoting the upregulation of epithelial markers and downregulation of mesenchymal markers [102].

The derivation of newborn chick iPSCs was reported to be facilitated by and exhibit higher efficiency due to the addition of molecular inhibitors such as 2i (CHIR99021-0.75 µM and PD0325901-0.25 µM) to reprogramming medium in combination with thiazovivin-0.25 µM [27]. The study also demonstrated that the withdrawal of basic fibroblast growth factor (b-FGF) affected iPSCs colony morphology and alkaline phosphatase (AP) activity, indicating a dependency on FGF stimulation [27]. Researchers also explored the response of chicken ESCs when the culture medium is supplemented with LIF (leukemia inhibitory factor) and b-FGF. LIF was shown to maintain the totipotent characteristics of these cells, while the addition of b-FGF promoted their proliferation [24,104].

Feeder cells play a complementary role in maintaining the undifferentiated state, preventing spontaneous differentiation, and promoting stem cell self-renewal and proliferation [104,105]. These supportive cells create a suitable microenvironment by supplying specific requirements such as essential cytokines, growth factors, and nutrients and facilitating cell-to-cell interactions [104,105,106]. Several types of feeder systems have been evaluated for chicken iPSC derivations, including ouabain-resistant (STO) fibroblasts, buffalo rat liver (BRL) cells, and mouse embryo fibroblasts (MEFs) [104]. While various culture surfaces have been studied based on the understanding of interactions between the extracellular matrix proteins (ECMPs) and cell adhesion molecules (CAMs), this is still an ongoing area of research [34,104,106]. The development of the best surface for iPSC culture using feeder cells is still underway, and the widespread implementation of feeder cells is hindered by challenges related to the need for consistency, scalability, reproducibility, and animal-product-free culture surfaces, as well as considerations of validation, reliability, and cost [106].

In conclusion, the search for reprogramming methods and enhancers is continuously evolving, with ongoing advancements in techniques, small molecules, and culture conditions. These advancements hold great potential for improving reprogramming efficiency and expanding the range of cells that can be successfully reprogrammed.

## 6. A Brief History of Avian iPSC Production

Following the groundbreaking discovery in 2006 in mice and humans [15,16], researchers embarked on exploring the application of iPSC technology in other species, including birds. In 2012, Lu et al. achieved a significant milestone by producing the first nonmammalian iPSCs. The primary avian iPSCs were derived from quail embryonic fibroblasts (QEFs) obtained from 11-day-old embryos. The reprogramming process involved the delivery of human reprogramming factors, namely *POU5F1*, *NANOG*, *SOX2*, *LIN28*, *KLF4*, and *C-MYC*, via lentiviral transduction [31]. Lu’s pioneering work opened the way for discussing future research on chick stem cell progress and prospects [13], highlighting the potential applications of avian iPSCs in various fields, including developmental biology, regenerative medicine, disease modeling, and conservation biology. 

A subsequent study demonstrated that the lentivirus transduction of human OSKM to invertebrate cells such as Drosophila and vertebrate cell species including zebrafish and birds (quail, zebra finch, and chicken) could generate iPSC-like cells with many of the characteristics of natural mammalian and bird stem cells. This finding established a relatively conserved reprogramming process across mammals, birds, and even invertebrates, highlighting the significance of pluripotency gene usage in iPSC derivation [22,28].

Right away, in 2014, iPSCs were generated from chicken embryonic fibroblasts (CEFs) obtained from Barred Rock and Black Australorp chicken breeds. Lentiviral delivery of six human stem cell genes, including *POU5F1*, *NANOG*, *SOX2*, *LIN28*, *KLF4*, and *c-MYC*, successfully culminated in somatic reprogramming. These chicken iPSCs (ciPSCs) exhibited endogenous pluripotent markers such as *Oct4* and *Ssea1*. However, these cells also presented the expression of CXCR4 and germ cell-defining proteins, e.g., CVH and DAZL, resembling in vivo-sourced PGCs. Moreover, after in vivo transplantation, these cells demonstrated the ability to migrate to the developing gonads in 15-day-old embryos, suggesting their potential role in avian gonadal development. This finding presents a novel strategy for conserving important genetic traits in endangered bird species [22].

Also in 2014, to evaluate the optimal conditions for reprogramming, culture, and differentiation of chicken iPSCs, a comprehensive investigation on cytokines and media supplements was conducted to identify the ideal combination. Various grow factors, including LIF, recombinant human (rh)IL6, (rh)ILR6a, (rh)IGF1, (rh)FGF2, recombinant mouse (rm)SCF, and inhibitors PD0325901, CHIR99021, and A83-01, were found to enhance iPSC colony proliferation and morphology. However, the challenge of determining the optimal media and necessary supplements to overcome the rapid senescence observed in conventional CESC and iPSC cultures remains unsolved [107].

The search for nonviral alternatives for reprogramming avian cells began as early as 2014, with the successful production of iPSCs from chicken embryonic fibroblasts using premade minicircle DNA containing the human reprogramming factors *POU5F1*, *SOX2*, *LIN28*, and *NANOG*, delivered by cationic lipidic strategies. This was the first instance of avian iPSCs being derived by a nonviral method. This discovery offers promising alternatives that mitigate the drawbacks associated with viral integrative delivery, providing a safer and more cost-effective approach for iPSC production and expanding their potential applications in various fields [85].

In the gap between 2014 and 2016, Lu and colleagues [30] developed a step-by-step protocol, providing valuable guidance, to produce iPSCs from embryonic quail cells using lentiviral vectors in the delivery of human transcription factors, including *POU5F1*, *SOX2*, *NANOG*, *LIN28*, *C-MYC*, and *KLF4*. 

By 2016, research on avian iPSCs took on a new perspective, considering iPSC technology in the context of viral infections. The Newcastle disease virus (NDV) is a highly destructive pathogen affecting poultry and wild birds. Following the induction of iPSCs, investigations were conducted to assess the interaction between NDV and avian iPSCs, with the aim of developing a selection method that could enhance the tolerance of these cells to NDV-induced cellular damage. However, it was observed that iPSCs were permissive to NDV infection and susceptible to virus-mediated cell death [108]. Furthermore, iPSCs are being explored as potential substrates to produce viral poultry vaccines, replacing embryonated eggs (Figure 1). This advancement offers advantages such as improved scalability, reduced variability, and better control over the production process. The chicken iPSC line, BA3, is particularly promising for vaccine production due to its favorable characteristics, including high cell density growth and the ability to grow in serum-free medium [109]. Three years later, a ciPS cell line capable of producing a replication-incompetent virus was successfully established. This achievement offers a new strategy for studying viral diseases and cell-based vaccine production. By inactivating the virus within the ciPS cells, safer and more efficient methods for vaccine development and disease prevention can be explored. Further research is needed to fully realize the potential of ciPS cells in virus research and vaccine production [110]. 

Choi et al., in 2016, shed light on the structural changes that occur within chicken cells during the process of reprogramming, providing insights into the cellular mechanisms involved in iPSC generation and pluripotency acquisition. Through electron microscopy, it was observed that intracytoplasmic organelles in differentiated somatic cells underwent remodeling during cell reprogramming and that ciPSCs exhibited a higher nucleus-to-cytoplasm ratio and contained globular mitochondria with immature cristae [111].

A significant step in avian iPSC production occurred in 2017 with the generation of the first iPSCs derived from nonembryonic cells. FFCs from adult chickens (older than 24 weeks) were successfully used as the cell source for iPSC production. Retroviral vectors, including mouse pMXs-*Oct3/4*, pMXs-*Sox2*, pMXs-*Klf4*, pMXs-*cMyc*, and pMXs-*Nanog*, were also utilized in the reprogramming process [24]. The utilization of adult cells presented an opportunity to overcome the significant obstacle of the impractical and potentially detrimental use of embryos from endangered avian species. 

Continuing in this line of research, in 2018, Katayama successfully generated iPSCs from muscle fibroblasts of chickens on day one after hatching. The study employed nonviral polycistronic reprogramming vectors, PB-6F and PB-R6F, which contained mouse M3O (a transactivation domain derived from MyoD fused with *Oct3/4*), *Sox2*, *Klf4*, *c-Myc*, *Lin28*, and *Nanog*. The transfection of the vector into the cells was performed using cationic lipids. To enhance the reprogramming efficiency, the culture medium was supplemented with factors such as LIF, FGF, thiazovivin, CHIR99021, and PD0325901. This approach contributes to expanding the repertoire of techniques used to generate avian iPSCs [35]. 

In 2018, it was further confirmed that the combination of the four transcription factors OSKM is sufficient for generating iPSCs in avian species. However, the addition of *Nanog* and *Lin28* proved to not only enhance the reprogramming efficiency but also overcome the senescence, crucial for the long-term culture and maintenance of these iPSC colonies. Notably, in specific avian species like ducks, it was demonstrated that the complete derivation of iPSCs heavily relied on the inclusion of *Nanog* and *Lin28* [56].

In an effort to optimize the culture conditions for avian iPSCs, various feeder cells were evaluated for the cultivation of chicken iPSCs. A study found that the most effective culture system for the growth and proliferation of ciPSCs includes utilizing MEF feeder cells combined with embryonic germ cell culture medium [104]. 

Among the most recent studies, in 2021, the use of iPSCs in generating germ cells for conservation purposes was reported. While it was already known that iPSCs could result in germ cell fate [22], Zhao et al. [23] showed that iPSCs derived from CEFs, transfected with pCDH-CMV-MCS-GFP vector carrying OSNL genes, successfully differentiate into PGCs. Following PGC induction, the research group still explored the allotransplantation of these cells into recipient embryos, resulting in viable offspring with characteristics inherited from the donor cells. This line of work seeks to cooperate in the preservation of genetic diversity in endangered avian species by reproductive chimera production using iPSC technology [23]. Additionally, the study highlighted the use of vitamin C and valproic acid as adjuvants to improve the efficiency of the reprogramming process.

Last but not least, iPSCs were successfully produced from endangered wild avian species, including the Okinawa rail, Japanese ptarmigan, and Blakiston’s fish owl. FFCs and dermal fibroblasts obtained immediately postmortem were used as cell sources. The cells were reprogrammed using the PB-TAD-7F transposon vector, which contained the mouse genes *Oct4*, *Sox2*, *Klf4*, *c-Myc*, *Nanog*, *Lin28*, and *Klf2* and was delivered with cationic lipids. The reprogramming process was optimized with the addition of LIF, FGF, thiazovivin, CHIR99021, and PD0325901. Furthermore, the authors convincingly demonstrated that the incorporation of the *Yap* gene during reprogramming was essential for the generation of Japanese golden eagle iPSCs. This achievement was accomplished through the utilization of the PB-DDR-8F reprogramming vector [27]. This underscores the importance of tailoring reprogramming strategies to meet the specific needs of different avian species, particularly those that are endangered. 

Amongst these findings, the significant milestones of iPSC research in avian species are chronologically summarized in the timeline depicted in Figure 3. In addition to the ongoing research on exploring minimally invasive somatic cell sources and developing biosafety protocols for reprogramming, further studies should be conducted to elucidate species-specific requirements for reprogramming, optimize the process, and achieve the production of high-quality iPSCs. These efforts are essential to advance the field and address the specific needs of different avian species. By understanding the unique characteristics and demands of each species, researchers can develop tailored reprogramming approaches that maximize the efficiency and quality of iPSC generation. Ultimately, these endeavors will contribute to the broader goal of utilizing iPSCs in avian conservation and various applications in avian biology research. 

## 7. Pluripotent Cellular Characterization and Findings

All studies related to iPSC derivation from either humans or animals utilized two or more methods to characterize and validate the cellular reprogramming. For that, they used morphological, molecular, or physiological analyses. Table 1 presents an overview of the methodologies employed by the main studies for validating the findings on avian iPSC generation.

When introducing exogenous genes into somatic cells, it is expected that these genes will be expressed and will progressively increase during the reprogramming process [28,49,51]. Consequently, in an RT-PCR analysis, their derivatives should show an increase in expression. Genes such as *Oct4*, *Nanog*, *Sox2*, *Sox3*, and the *Klf* family can be used as pluripotent markers and to assess the translation process and the expression of proteins for pluripotency, such as TRA-1, SSEA4-SOX2, NANOG, and OCT4; immunocytochemistry techniques are commonly performed [22,24,27,28,35,51,85,107,111]. Once reprogrammed, the cellular metabolism shifts [52,55,60,62]. An accessible metabolic marker for the pluripotent state is the activated alkaline phosphatase in iPSCs colonies [101,112]. In a fully differentiated iPSC, functional pluripotency is expected. This means that these cells should be able to differentiate in teratomas in vivo consisting of all three primary germ layers [113] or form an embryoid body (EB). One important but no so common validation is the maintenance of the normal karyotype. This guarantees the cell genomic stability compatible with the target species.

Overall, studies in birds employed multiple approaches to characterize iPSCs and validate their reprogramming potential [114]. The findings support the pluripotent nature of iPSCs and their ability to differentiate into various cell lineages, highlighting their potential applications in avian conservation [12,13].

## 8. Attention Points in the Use of iPSCs

The first clinical trial using hiPSCs took place in 2014, and since then, different groups have evaluated the medical use of these cells [115]. Despite the significant promise that iPSCs hold, challenges to their application persist. The presence of (epi)genetic instability and variations raises concerns about the safety of utilizing these cells, as the potential adverse effects of identified mutations remain uncertain [116,117].

Among the most common genetic alterations reported in mammalian iPSCs is the trisomy of chromosome 12, housing genes related to cell cycle regulation and the pluripotency-associated gene Nanog [118,119,120,121]. Also frequent is the amplification of chromosomal region 20q11.21, which harbors antiapoptotic genes like *BCL-XL****/****BCL2L1*, as well as duplications of oncogenes and deletions of tumor-suppressor genes [122,123,124]. These genomic mutations are usually associated with tumorigenicity, favoring the survival and expansion of abnormal cells and increasing the chances of selecting cells with these alterations [125,126]. This can impact the in vivo applicability of iPSCs and their reliability as in vitro study models.

Genetic instability in iPSCs can arise from various factors. Approximately fifty percent of the mutations identified in hiPSCs were present at low levels in their parental cells. The preexisting variations exhibit correlations with specific somatic sources and the age of the donor, with older animals and elderly patients tending to yield iPSCs characterized by a heightened mutational number [127]. Genetic variations can also occur during the reprogramming stage, associated with the reprogramming method wherein integrative factors like viral vectors may induce insertion mutations [128]. Furthermore, these variations can manifest during the passage process and even during cell differentiation [122,125].

It is essential to emphasize that iPSCs are not directly employed in transplantation procedures due to their pluripotent nature, which could lead to teratoma formation. Consequently, these cells undergo in vitro differentiation into target cells for use in vivo. Thus, in addition to the valid concern regarding the presence of residual undifferentiated cells in transplanted populations capable of forming tumors, the investigation of genomic mutations associated with tumorigenicity in these cells becomes paramount [117].

Therefore, genetic abnormalities should be traced through each stage of this cell preparation, from reprogramming to differentiation. While Giemsa G-banding can be used to detect numerical (aneuploidy and polyploidy) or large structural chromosomal changes [129], single nucleotide polymorphism (SNP) arrays are important to detect the variation of a unique nucleotide through the whole genome [130]. 

Studies with mammalian cells have demonstrated that epigenetic changes normally precede genetic alterations [131]. Somatic cells have a more compacted chromatin compared to stem cells [132]. The reprogramming process involves altering the epigenetic landscape, encompassing DNA methylation and histone tail modifications (acetylation and methylation), to activate pluripotency-associated genes and silence somatic cell-specific genes. Low-passage iPSCs or those undergoing incomplete reprogramming may display differential methylation regions (DMRs) that retain residual DNA methylation signatures characteristic of their somatic tissue of origin, indicating an incomplete epigenetic reset [133,134,135,136]. This “epigenetic memory” can influence the potential for lineage differentiation, favoring differentiation along pathways related to the donor cell while simultaneously limiting alternative cell fates and impacting the biological function of iPSC lines [134]. Consequently, the consideration of DNA methylation profiles becomes imperative when assessing iPSC lines for applications such as cell therapy. As of now, the mechanisms that control the generation of fully pluripotent avian iPSCs have not been sufficiently investigated, and no studies have been undertaken to characterize the (epi)genetic variations in avian iPSCs. More research into understanding iPSC genome stability needs to be explored in order to use these cells to aid in wildlife preservation and restoration.

## 9. Application and Future Perspectives

The technology of iPSCs in avian species is relatively new, particularly concerning threatened wild bird species. Considering the diverse demands of different exotic species, it is crucial to understand the specific requirements of somatic acquisition and iPSC derivation protocols for each species to ensure the production of high-quality iPSCs. 

The search for alternative somatic cell sources from tissues obtained through minimally invasive collection and the development of biosafe yet highly efficient reprogramming methods should continue to be prioritized. The utilization of iPSCs would allow for increased genetic diversity among populations through crossbreeding between conservation sites and zoological institutions. For that, the creation of biobanks with collectible tissues worldwide for genetic reserve purposes is of the utmost importance [137]. Biobanks facilitate the preservation and storage of genetic material, enabling breeding programs once protocols have been well established.

These advancements hold great potential for the establishment of conservation projects. It is true that the in vitro gametogenesis (IVG) process remains immature, with limited studies conducted in mammals and no exploration in the avian domain. The successful production of fully grown oocytes in mice [138], along with the acquisition of oogonia and immediate precursor cells for human oocytes [139], represents a significant advancement in reproductive science. However, much remains to be learned and standardized in this field. Despite the absence of expertise in generating gametes in vitro, the actuality of reproductive chimera production through xenotransplantation of PGCs derived from iPSCs is already established (Figure 1) [10,26]. 

The use of iPSCs can extend beyond conservation of avian endangered species. The food industry holds big expectations regarding the sustainably cultivated meat revolution. Considering meat as a multitissue structure, domestic bird iPSCs could be used to derive somatic muscle tissue and adipogenic and hematopoietic tissue [106,107,108]. In addition to enabling animal-based production without harming animals, iPSCs are presented as an eco-friendlier commercial possibility, as they alleviate the environmental burden that farm animals place on the environment (deforestation, water consumption) [140]. 

Complementarily, iPSCs or their derived PGCs can be subjected to gene editing to increase production or decrease their components in chicken meat or eggs. For example, albumin-free eggs could be acquired and used for vaccine development [109]. Also related to vaccine production, iPSCs can be used in toxicology studies and the understanding of pathogenic mechanisms, disease resistance, and vaccine development. This highlights the versatile applications of avian iPSCs in addressing conservation, agricultural, and medical needs (Figure 1).

Finally, further research is necessary to explore and optimize iPSC protocols in avian species, with a focus on species-specific requirements. Through the establishment of high-quality iPSCs and the development of effective methodologies, iPSCs have the potential to revolutionize avian conservation efforts and contribute to the broader field of biology.

## 10. Conclusions

iPSCs emerge as a promising avenue of exploration across various realms of biological study. Their significance extends notably to the conservation efforts targeting endangered avian species. The intricate process involved in iPSC derivation demands meticulous attention and active decision making at each stage. Understanding that the quality of iPSCs is contingent upon the somatic cells from which they originate, careful selection of the appropriate cellular source is imperative. Furthermore, unexplored somatic cell sources, such as peripheral blood mononuclear cells (PBMCs), hold promise and merit investigation, especially with a focus on developing less invasive methods for cellular collection for avian species. Another crucial consideration is the specificity of the combination of reprogramming factors. The roles of the four Yamanaka factors appear to be conserved among vertebrates, but there are subtle requirements to enhance the efficiency and quality of cellular reprogramming. Thus, adapting protocols to the specific demands of avian species is essential, including adjustments in culture conditions, medium formulations, and the application of enhancers at defined intervals. Gene delivery is also a critical step that warrants attention. While various validated alternatives, such as viral vectors (e.g., retroviruses and lentiviruses), exist, ongoing research seeks to identify equally efficient alternatives that result in a low incidence of DNA sequence variations in iPSC lines. Challenges in the in vivo use of iPSCs still persist, demanding due diligence. Incomplete reprogramming, epigenetic memory, unstable genotypes, and epigenetic anomalies may manifest. Thus, rigorous validations through methods such as karyotyping, gene sequencing, and nucleotide-level mutation analysis are necessary to mitigate these issues. This is particularly relevant in the conservation of endangered species, where maintaining genome integrity is imperative for successful reproduction and the reintroduction of specimens into natural habitats. 

Overall, while iPSC technology is a relatively recent development, its application in avian species is even more nascent. Substantial progress has been made for avian iPSCs; however, numerous knowledge gaps remain, necessitating exploration into avian cellular and molecular biology nuances. Undoubtedly, iPSCs hold immense potential not only in the conservation of endangered wildlife, establishment of biobanks, and reproductive programs but also in fields such as food production, pharmacology, and healthcare. As this technology continues to evolve, addressing existing challenges will be imperative to realize the full spectrum of benefits iPSCs offer to diverse scientific domains.

## Figures and Tables

**Figure 1 animals-14-00220-f001:**
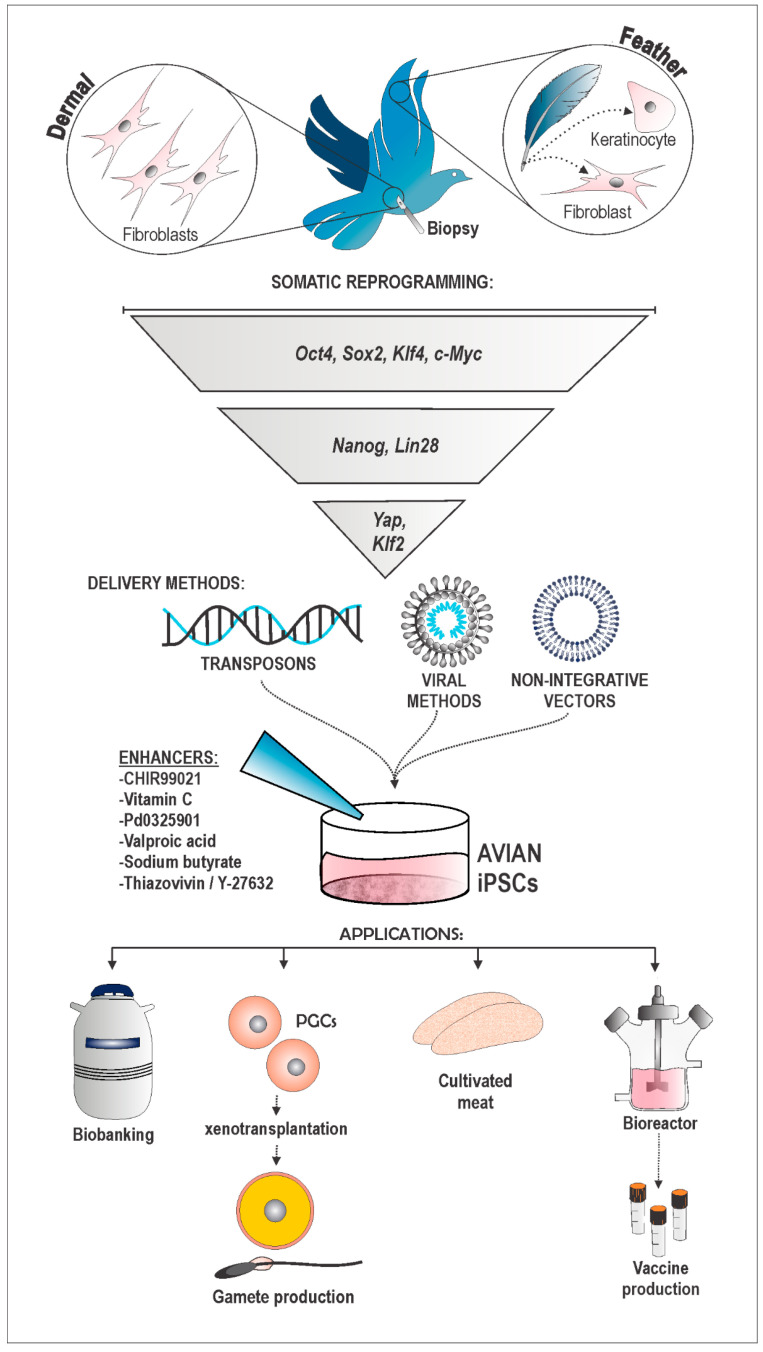
Schematic drawing depicting the process of avian iPSC generation. The scheme delineates the workflow employed in general by various research cohorts to successfully establish the derivation of avian iPSCs. The process hinges upon the noninvasive harvesting of somatic cells, specifically dermal fibroblasts, FFCs, or keratinocytes. Somatic reprogramming into a fully pluripotent state has been accomplished through diverse combinations of the OSKMLNKY gene cocktail. Various delivery methods, including viral vectors, transposons, and minicircles, have been utilized to introduce reprogramming genes into avian cells. The diagram further highlights the predominant enhancers employed in cell culture media. Ultimately, potential applications of iPSCs are outlined, encompassing conservation efforts through germline derivation, establishment of biobanks, and utilization in biotechnological domains such as vaccine production and cellular agriculture.

**Figure 2 animals-14-00220-f002:**
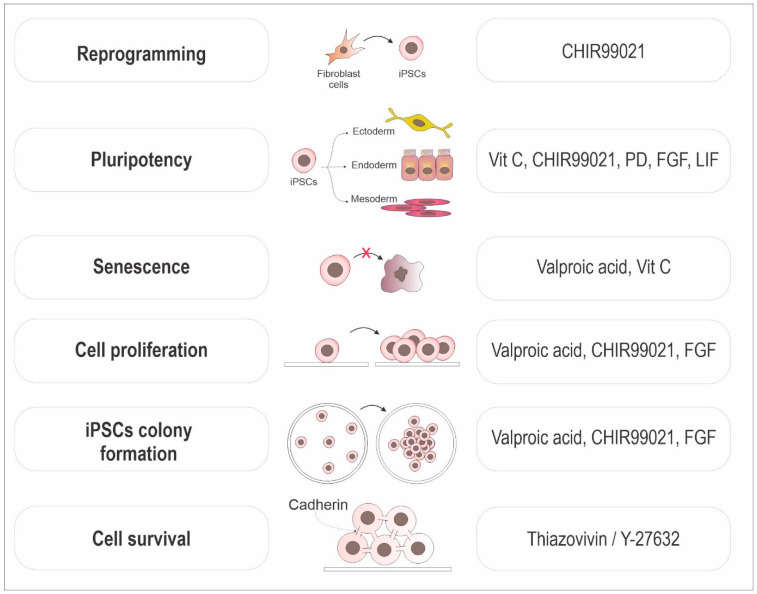
Mechanism of action of iPSC production enhancers. Schematic model grouped by the effects promoted by enhancers to optimize the process of inducing pluripotency and cell viability. A single enhancer can act to favor different mechanisms.

**Figure 3 animals-14-00220-f003:**
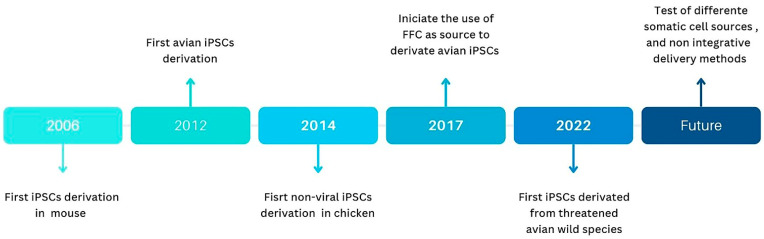
Avian iPSC research milestones timeline. The timeline includes the first avian iPSCs ever generated, the primary avian derivation without using nonviral reprogramming methods, the pioneer iPSCs independent of an embryonic cell source, and the first iPSCs for endangered avian species.

**Table 1 animals-14-00220-t001:** Overview of methods employed by main studies for characterizing and validating avian iPSC generation.

Reference	[27]	[23]	[35]	[24]	[30]	[85]	[28]	[31]
Cellular source	FFCs, dermal fibroblasts	CEFs	Dermal fibroblasts	FFCs	QEFs	CEFs	CEFs, QEFs	QEFs
Reprogramming method	PiggyBac transposon vector	Lentiviral vector	PiggyBac transposon	Retroviral vector	Lentiviral vector	Minicircle DNA	Lentiviral vector	Lentiviral vector
Reprogramming genes	*Oct3/4*, *Sox2*, *Klf4*, *c-Myc*, *Lin28*, *Nanog*, *Klf2*, *Yap*	*Oct3/4*, *Sox2*, *Lin28*, *Nanog*	*Oct3/4*, *Sox2*, *Klf4*, *c-Myc*, *Lin28*, *Nanog*	*Oct3/4*, *Sox2*, *Klf4*, *cMyc*, *Nanog*	*OCT3/4*, *SOX*, *KLF4*, *C-MYC*, *LIN28*, *NANOG*	*OCT3/4*, *SOX2*, *LIN28*, *NANOG*	*Oct3/4*, *Sox2*, *Klf4*, *c-Myc*	*OCT3/4*, *SOX2*, *KLF4*, *C-MYC*, *LIN28*, *NANOG*
Alkaline phosphatase	Yes	Yes	Yes	Yes	Yes	Yes	Yes	Yes
Immunocytochemistry (*Pluripotency markers*)	SSEA-1 SSEA-3 SSEA-4	SSEA-1	SSEA-1 SSEA-3 SSEA-4	SSEA-1	POUF SSEA-1	POU5F1 SSEA-1 SSEA-4 SOX2 NANOG	SSEA-1	POU5F1 SOX2 SOX17 SSEA4 TRA-1-60 TRA-1-81
RT-PCR *(Pluripotency-associated genes)*	*DNMT3B* *ESRRB* *FBXO15* *LIN28* *NANOG* *chicken PouV* *REX1* *SALL4* *SOX2* *SOX3*	*KLF4* *LIN28* *NANOG* *POU5F1* *SOX2* *SSEA1*	*ESRRB* *KLF2* *cKLF4* *chicken Nanog* *chicken PouV* *REX1* *cSOX2* *TBX3*	*c-MYC* *KLF4* *LIN28* *NANOG* *POU5F1* *chicken PouV* *SOX2*	-	*DNMT3B* *LIN28* *NANOG* *POU5F1* *chicken PouV* *REX-1* *SLC2A3* *chicken Sox2* *SOX2* *TERF1*	*KLF4* *c-MYC* *NANOG* *POU5F1* *SOX2*	*POU5F1*
Telomerase activity	-	-	-	-	-	Yes	Yes	Yes
Embryoid body formation	Yes	-	Yes	Yes	-	Yes		Yes
In vitro differentiation	Neural and smooth muscle		Neural cells	Endo-, meso-, and ectoderm	-	Endo-, meso-, and ectoderm		Endo-, meso-, and ectoderm—neural
Teratoma formation	Yes		Yes	-	-	-	Yes	-
Proliferation assay	-		-	-	-	Yes		Yes
Production of chimera	Yes	Yes	-	Yes	Yes	Yes	Yes	Yes
Karyotype	Yes		Yes	-	-	-	Yes	-
Other analysis	Rnaseq, mitochondrial staining	Rnaseq, bissulfete sequencies, DNA methylation, flow cytometry, short tandem, Western blot	Rnaseq	Mitochondrial staining	-	Flow cytometry	-	-

## Data Availability

Data sharing not applicable.

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
