# Peer review of "Avian iPSC Derivation to Recover Threatened Wild Species: A Comprehensive Review in Light of Well-Established Protocols"

_animals, 2024, doi:10.3390/ani14020220_

Round 1

Reviewer 1 Report

Comments and Suggestions for Authors

This manuscript, entitled „Avian iPSC Derivation to Recover Threatened Wild Species: A Comprehensive Review in the Light of Well-stablished Protocols”  written by Iara Pastor Martins Nogueira and co-authors present the review of the up-to date iPSC derivation methodology in term of using it in the conservation of endangered avian species. The authors referred to the recent and relevant publications, well reviewed the field of reprogramming, the source of the cells, the methodology. However it would be valuable for the manuscript to highlight the differences between mammalian and avian cells - if there are any in terms of culture methods (besides the source of the cells) or cell biology, which can affect reprogramming procedure.

From this review, it is very interesting that YAP commonly promotes the derivation of iPSC from avian cells, it is worth mentioning that the function of YAP is tricky and can be different in human and mouse cells and that YAP/TAZ can also act differently depending on the stage of cell/organism development. Maybe it is worth mentioning that it would be very interesting to further study the YAP as the avian reprogramming factor, but not so common in the reprogramming of human or mouse cells.

Comments on the Quality of English Language

I have some other questions and comments, listed below:

1.      Line 64: “…dilemmas associated with using ESCs.. “ or  “related to the use of ESCs”.

2.      The abbreviation for CEFs can be introduced in line 112 instead of 568

3.      Line 188 and next: …keratinocyte…exhibit characteristics of multipotent-like stem cells. These keratinocytes, located at the base of the internal epidermal layer of the feather follicle can be segregated from the FFC fibroblasts…

4.      Line 196: “Besides keratinocytes use for hiPSCs derivation, urine cells offer a non-invasive means of collection that does not require professional human assistance, making…”

5.      Line 204: “from urine cells may be…

6.      Line 205: sentence  is not clear: …”would depend on the scientist at the right”

7.      Line 215: instead “iPSC induction” authors can change for “iPSC derivation”

8.      Line 225: …”OSKM is capable of deriving iPSC”

9.      Line 231: the sentence is not clear, can be replaced for “..it is not autoregulated and plays a distinct role in the metabolic processes of cells undergoing reprogramming.…”

10.   Line 258: it would be easier for readers, that in this sentence, before the abbreviations with “h”, like hPOU5F1, hNANOG…authors can insert the  …“fibroblasts transduced with six human reprogramming factors…”. And in whole text it should be clearly explained the abbreviation of the letter c or h(for chicken or human? ) for example in the table 1 “cLIN28, cNANOG” ect.

11.   Line 304: …”when considering their potential use in breeding programs…”

12.   Line 310, the sentence is not clear: .”lentiviral vectors are still cannot considered safe for therapeutic…”

13.   Line 344: should be “transposons”

14.   Line 431: “…and MEK pathway, can improves reprogramming efficiency…”

15.   Line 441: “…[27]. This study…”

16.   Line 498 : please explain the abbreviations for rhIL6, rhsILR6a.. it can be written “recombinant human ....”

17.   Line 541: “…iPSCs from non-embryonic cells. FFCs…”

18.   Line 604: “Pluripotent Cellular Characterization  And Findings”

19.   Line 624: “target species.”

20.   Line 637 “the development of biosafety yet …”

21.   Table 1: the headings of the rows should fit the width of the column (to avoid the splitting of the words: “Reprogrammi ng method” ect..).  In reference 31, RT-PCR “hPOU5F1” should be italic. In this row(RT-PCR) the POU5F1 is inconsistently  written: POUV, cPOU5, cPOUV .  The heading “Others” maybe should be replaced for “Other analysis”

22.   The reference [23] is a retracted article.

Reviewer 2 Report

Comments and Suggestions for Authors

Reviewer 3 Report

Comments and Suggestions for Authors

Dear editor,

The review is focused on the avian iPSC derivation. The topic is interesting and important in the light of pluripotency studies and threatened species conservation.

There are several major issues that need to be addressed.

1. English native speaker should correct numerous mistakes.

2. There shall be a chapter about problems with the iPSC technology. The authors describe iPSCs as the perfect solution to species conservation, and it is clearly not the case. Among the problems that should be addressed are at least those: 1) pluripotent stem cells are known to have unstable genotype compared to somatic cells. 2) There is epigenetic instability. 3) Incomplete reprogramming resulting in “epigenetic memory” of the initial cell type. 4) In vitro selection of cells with increased number of protooncogenes (such cells divide faster). 5) Lack of technology for gametogenesis in vitro, there are just a few articles for mice and human iPSCs.

3. The review is divided in 8 chapters but inside the chapters there is lack of logic in the description of the subject. The logical structure should be improved, may be through additional chapters/subchapters.

Overall, the review shall be extensively rewritten and not ready for the publication.

Minor remarks:

In several cases genes are written not in italics.

L71. “iPSCs allows access to the entire genetic repertoire” – that is impossible as it required collection of the genetic material from all specimens of the population.

P3. Figure 1. Letters V and Y are indistinguishable; it is important as there is YAP gene.

L119. Animal MSC do not have germinative lineage potential. It is theoretically possible that cMSCs can differentiate it that but most probably the reference is not trustworthy. What is known about MSCs, they can merge with various cell types. Earlier reports about “pluripotency” of MSCs are now explained by cell fusion, not differentiation.

L143. What is the meaning of the “resistance” of fibroblasts?

L295-311. It is true that introduction of retroviruses into the genome is not a good idea. On the other hand, in mammalian pluripotent stem cells retroviruses are silenced. Despite the occasional reactivation retroviral vectors still present a viable though not ideal option. That should be mentioned. And, as for avian species, is there inactivation of retroviral transgenes in iPSCs?

L337-338. Please provide references to the efficiency of retroviral transduction into avian cells.

L401. As far as I remember, reference 79 was never reproduced and is a sham or simply bad science. Some comments shall be included if this reference is cited.

L427. Thiazovivin is not the only ROCK inhibitor. In human iPSCs Y-27632 is widely used.

P11. Figure 2. Cadherin, not Caderin.

Comments on the Quality of English Language

English native speaker should correct numerous mistakes.

Round 2

Reviewer 3 Report

Comments and Suggestions for Authors

Dear editor,

The authors adressed my previous suggestions. There are some punctuation mistakes, in some instances gene names are not in italics, otherwise I do not see major issues.

The MS may be published.

Comments on the Quality of English Language

There are some punctuation mistakes, in some instances gene names are not in italics.

Author Response

Reviewer #3

Comments and Suggestions for Authors

The authors addressed my previous suggestions. There are some punctuation mistakes, in some instances gene names are not in italics, otherwise I do not see major issues.

The MS may be published.

Response: We express our appreciation for the comments received. Indeed, there were some punctuation errors that have been rectified. The gene nomenclature has all been thoroughly reviewed and formatted to italics for those that were not already in accordance. All modifications have been highlighted in red to facilitate identification.